# Update on Domestic Violence and Traumatic Brain Injury: A Narrative Review

**DOI:** 10.3390/brainsci12010122

**Published:** 2022-01-17

**Authors:** Kellianne Costello, Brian D. Greenwald

**Affiliations:** 1Rutgers Robert Wood Johnson Medical School, Piscataway, NJ 08854, USA; kc1113@rwjms.rutgers.edu; 2Department of Physical Medicine and Rehabilitation, Rutgers Robert Wood Johnson Medical School, Piscataway, NJ 08854, USA; 3Hackensack Median Health-JFK Johnson Rehabilitation Institute, Edison, NJ 08820, USA; 4Department of Physical Medicine and Rehabilitation, Hackensack Meridian School of Medicine, Nutley, NJ 07110, USA

**Keywords:** traumatic brain injury, domestic violence, intimate partner violence, post-traumatic stress disorder

## Abstract

Research on traumatic brain injury (TBI) as a result of domestic violence has greatly increased in the past decade, with publications addressing the prevalence, diagnosis, evaluation, and treatment. Although TBI due to domestic violence has recently been found to occur quite frequently, it was not widely understood until the 1990s. Individuals who suffer from domestic violence TBI often experience sequelae such as decreased cognitive functioning, memory loss, and PTSD. The goal of this article is to increase awareness about TBI secondary to domestic violence, with the intent that it will highlight areas for future research on the diagnosis, evaluation, and treatment of TBI in this population. The articles in this study were first found using the search terms traumatic brain injury and domestic violence. Although, in recent years, there has been a significant increase in research on TBI due to domestic violence, the overall conclusion of this review article is that there is still a need for future research in many areas including the effects on minority populations, the effects of COVID-19, and improvements of screening tools.

## 1. Introduction

Traumatic brain injury (TBI), which is the leading cause of all deaths for Americans less than 44 years old, is defined as “an alteration in brain function, or other evidence of brain pathology, caused by an external force that may result in cognitive impairment” [1,2]. TBIs are categorized by the Glasgow Coma Scale, with scores of 13–15 being mild, 9–12 being moderate, and less than 8 being severe TBI [2]. Despite 80% of all reported TBIs being mild, the incidence of mild TBIs (mTBI) is very underreported [1], likely because mTBI can present as a transient feeling of being dazed, disoriented, or confused [3]. Without proper treatment, there is a risk for repetitive injuries which can have a cumulative result, leading to detrimental outcomes such as chronic traumatic encephalopathy, depression, suicidality, and Alzheimer’s-like syndromes [2].

Intimate partner violence (IPV) is a type of domestic violence that is defined as “behaviors that are intended to exert power and control over another individual, including physical, sexual, verbal, emotional, and financial abuse, and/or stalking” [4]. The detrimental outcomes of intimate partner violence cost about 5–10 billion dollars a year [5]. IPV is the most common type of violence experienced by women, with approximately one in three women experiencing IPV in their lifetime [2]. Though TBI secondary to domestic violence is often seen in women, it is important to note that men can also be victims of IPV, with studies showing that about 14% of men experience IPV in their lifetime [6].

Until the recent past, there has been a dearth of research on the effects of TBI on women, and even less research done on TBI due to domestic violence. Prior to the 1970s, domestic violence was often laughed about in plays and pop culture with jokes being made such as “plenty of wives take more punches than boxers do and bury several husbands”. In addition, before 1990, TBI research primarily focused on males, since only male animals were used for research because the reproductive cycle of female animals affected the results [7]. Battered woman syndrome was first described in 1975, at which point it was described as more of a social and psychological issue than a medical issue. Since then, research on TBI due to domestic violence has increased [8]. It has now been estimated that the number of women who have experienced TBI secondary to domestic violence is 11–12 times greater than the number of TBIs experienced by military personnel and athletes combined [9]. Domestic violence can lead to TBI through aggressive shaking, strangulation, a blow to the head (with a fist and/or heavy object), and falling/being thrown to the floor [10]. In addition, 25% of overall TBIs are caused by violence, many of which have mild symptoms [11].

## 2. Materials and Methods

References for this review were obtained using a search of online databases: PubMed, Medline, Cochrane, CINAHL, and Academic Search Premier (Figure 1). The search was conducted using the following key terms: traumatic brain injury and domestic violence. Boolean operator “AND” was used to connect search terms and narrow results. Truncations and MeSH headings were not used. Some references were not identified using the online databases but were obtained through reference lists of other articles. Articles before the year 2010 were excluded from the search unless the research was important for this paper and was not covered in more recent research. Only papers printed in or translated into English were included. Only papers with human subjects were included. We excluded case reports as sources. Studies on pediatric populations were not included. Our inclusion criteria were journal articles that looked primarily at traumatic brain injuries due to domestic violence. Please see Figure 1 for specifics about the article selection process. The original search did not include articles addressing the effects of domestic violence on the lesbian, gay, bisexual, transgender, queer, intersex, asexual (LGBTQIA+) communities or the effects of COVID-19 on the incidence of domestic violence. Separate searches were done on PubMed to find recent articles addressing these topics. These five articles are discussed in the sections labeled “Domestic Violence in the LGBTQIA+ Community” and “Effects of COVID-19 on Domestic Violence”, respectively.

## 3. Prevalence and Demographics of TBI Related to Domestic Violence

There are many factors that contribute to an individual’s risk of domestic violence (Table 1); therefore, it is important to keep in mind that anybody could be affected by domestic violence. Domestic violence is not limited to any demographic, so it is important that providers do not make assumptions about a patient based on stereotypes. Those at the greatest risk of domestic abuse are females of child-bearing age [12]. It is important to remember that adolescents can also be victims of intimate partner violence, since this population has an increased risk of suicidal ideation [13]. Individuals with disabilities are also at an increased risk of experiencing IPV [12]. One article, in particular, found that men with disabilities are at an increased risk of experiencing intimate partner violence as compared with women (both disabled and not disabled), as well as men who are not disabled [14].

TBI and domestic violence function as risk factors for each other. Those who receive TBI from domestic abuse and do not receive treatment often suffer from cognitive dysfunction, memory loss, and mood irregularities [2,3,15,16,17,18]. These challenges can lead to further frustration in the perpetrator, resulting in the occurrence of repetitive abuse that can lead to repetitive injuries to the head.

### 3.1. Effects of Socioeconomic Status

The World Health Organization (WHO) has found that the highest predictor of domestic violence is a low annual household income. In addition, a high socioeconomic status and high education level are protective against domestic abuse [24]. This was confirmed in another survey that showed 60% of women who lived in a rural environment experienced domestic violence, while only 40% of women who lived in an urban environment experienced domestic violence [11].

### 3.2. Effect of Race

Among minority groups, it has been found that domestic violence was associated with negative physical health outcomes such as disordered eating patterns, genitourinary dysfunction (vaginal discharge, burning during urination, and menstrual irregularity), as well as increased sexual risk taking that led to a higher likelihood of HIV infection [25].

Native American and Black persons are more likely than White people to be victims of violent TBIs [2,11,19]. The highest rate of violence among racial groups is that of Native American women, who have a three times higher risk of violent TBI and blunt-force trauma than White women [11]. Black women experience increased rates and severity of domestic violence, including a higher risk of having weapons used against them. This increased prevalence of severe domestic violence leads to higher rates of intimate partner homicide, and an overall greater mental health burden among Black women [26].

Immigrant Asian women, particularly Asian women who are sponsored by non-Asian men for marriage, also have an increased risk of TBI secondary to domestic violence. Oftentimes, these women are subject to severe violence for longer periods of time because they are isolated from their families and friends who live in a different country. In addition, in other countries, violence against women is more common and more widely tolerated, therefore, these women may be more likely to not seek help. It has also been seen that, in these relationships, the husbands’ family members, such as their mothers, may encourage their sons to be abusive towards their immigrant wives [20].

While individuals of all races can be victims of domestic violence; studies show that minority women are at an increased risk of violence, both in incidence and severity. Further research is needed in all of these populations, so that they can be better identified and treated.

### 3.3. Veterans

Female military veterans are 1.6 times more likely to experience domestic violence throughout their lifetime as compared with women who are not veterans [21,22]. These women are more likely to be abused during their time in the military as well as after. Due to their time in the military and their increased risk of abuse, they are also more likely to have mental health conditions including a diagnosis of substance abuse. It is important that providers spend time with these patients when diagnosing, because their mental health symptoms could easily be attributed to not only their time in the military [22]. This can be detrimental to the patient, as many of them need treatment that is specific to survivors of domestic violence.

### 3.4. Domestic Violence in the LGBTQIA+ Communities

The lesbian, gay, bisexual, queer, intersex, and asexual (LGBTQIA+) communities suffer from a higher rate of IPV as compared with their heterosexual and cisgender counterparts. In addition, individuals in these communities are also people of color, immigrants, and/or disabled are at the highest risk of IPV. Research has shown that minority stress in these groups can lead to increased IPV in their relationships due to struggles with discrimination and violence outside of the relationship. Individuals in these relationships may also battle with their own dislike and internalized hatred of their sexuality and/or gender identity [27].

Screening in these communities is even more important than in the heterosexual and cisgender communities, as this population is even less likely to report violence in their relationships, due to stories of discrimination of LGBTQIA+ community members who have reported IPV. These communities, especially trans individuals, are at an increased risk of mistreatment and assault by police. There have also been instances when healthcare personnel have not believed individuals when they report violence because they do not believe that it can exist between same-sex partners. There are also a number of shelters that will not accept trans individuals and other members of the LGBTQIA+ communities, making it extremely difficult for them to find a safe place to stay. It is important for healthcare providers to create an environment of acceptance and inclusion regarding this population, including visuals on websites and in the office waiting room. Patient paperwork should also be modified to be inclusive of the LGBTQIA+ communities [27].

Current measurement methods that have been utilized for assessing domestic violence typically have not included the LGTBQIA+ communities. Recently, new measurement tools have been developed to better measure the prevalence of domestic violence in this population. These new tools include the sexual and gender minorities-specific IPV Conflict Tactics Scale, the IPV Gay and Bisexual Men (-GBM) scale, and the transgender-related IPV Tool (Table 2) [27]. Though there has been some research done on domestic violence in the LGBTQIA+ communities, it is sparse and there is a need for more. There is also a need for future research on domestic violence in the LGBTQIA+ communities and its connection to TBI.

### 3.5. Domestic Violence in the Elderly and Children

Domestic violence is not limited to the abuse of intimate partners and can also include abuse of the elderly and children who live in the home. Elder abuse is defined as physical, psychological, financial, and social effects of individuals over 60 years old [13]. There is little research done on elder abuse, so it is a topic that needs to be explored in the future.

Infants are six times more likely to be abused than children aged 1–5 years old [23]. Children who grow up with instability and abuse are likely to suffer psychological, physical, and cognitive effects that lead them to be significantly more likely to be involved in domestic violence as adults [19]. It has been shown that 61.4% of patients who were abused as children went on to be abused as adults [17]. It has been found that those who experienced childhood instability are more likely to meet a significant other who also experienced abuse, leading to an abuser–abusee relationship [19]. It is important for children of abuse to receive proper treatment as children, including screening and care for TBI, to improve their chances of a successful adulthood.

## 4. Diagnosis of TBI Secondary to Domestic Violence

Whenever possible, victims of TBI should be recognized before they get to a healthcare facility. Oftentimes, the first person who meets a victim of TBI is a police officer who has been called to the house for a domestic violence dispute. Unfortunately, when police officers see victims of domestic violence, it is not always apparent to them that these individuals may need medical care. TBI symptoms can resemble the effects of drugs and alcohol, often leading to officers deeming these victims as noncompliant, and not recognizing that they need help [31]. Even in instances in which police officers identify that an individual may have a brain injury, it has been found that they are unlikely to intervene. Officers have said this is because they feel as though they are undereducated and underequipped to deal with these situations, making them afraid they will do more harm than good if they intervene [32].

Although police officers are not necessarily members of the healthcare field, as the first point of contact it is important for them to properly direct these individuals toward help. Patients have stated that they often feel that they were dismissed and blamed by police officers, which can lead to a stigma that makes them less likely to seek help from other resources [33]. It has been shown that domestic violence training programs have been successful in improving police officers’ understanding of domestic violence, and that it increased the chance of a perpetrator being arrested and convicted [31]. By educating officers on TBI secondary to domestic violence, it increases the chances that victims will be able to get to a healthcare facility where they can be properly screened and diagnosed.

### 4.1. Common Presenting Symptoms

It has been reported that 72% of domestic violence victims were not identified when they presented to the emergency department [34]. This is likely because there is rarely professional training on TBI secondary to domestic violence for front-line workers, and it can be difficult to identify symptoms of TBI secondary to domestic violence [35,36]. Many patients do not have obvious external injuries, and even if they do, many of these injuries could be attributed to other causes. Moreover, many patients may wait to seek care until years after the violence occurs, making it difficult to identify the connection between presenting symptoms and a patient’s history of abuse [6]. With all that said, there are several symptoms that are common among women experiencing TBI because of domestic violence. These symptoms include malnutrition, alcoholic cirrhosis, acquired thrombocytopenia, and post-traumatic wound infection [37]. It is important that healthcare workers can identify these symptoms and associate them with a possible result of abuse, indicating that the patient may need to be screened for TBI.

Women who have experienced domestic violence report poor overall health as compared with their counterparts that have not experienced abuse [21]. Physical symptoms that these women may present with include headache, back pain, loss of appetite, and abdominal pain. While TBI often results from physical violence, these women may also be experiencing sexual abuse which could lead to symptoms such as vaginal infections, STDs, vaginal bleeding, pelvic pain, and UTI [38]. Women who have been strangled may present with hoarseness, which can often be mistakenly attributed to screaming during an argument with their partner [39,40]. Patients who may have experienced strangulation should also be examined for petechiae in the conjunctiva, scalp, and external ear canal. If a patient is likely experiencing abuse, it is important to know that subconjunctival hemorrhages can present similarly to pink eye, so that one does not misdiagnose the patient [40]. It is particularly important to do a thorough exam on patients who have darker skin color, as bruises and other injuries could be masked by their darker complexion [26]. Patients presenting with TBI secondary to domestic violence may be emotionally labile, restless, combative, or even catatonic [40]. It is imperative that these personality characteristics are not just diagnosed as psychiatric because they could be indicative of a TBI that needs proper care (Table 3) [41].

### 4.2. Screening Tools

Victims of domestic violence are unlikely to report their abuse, with male victims being even less likely than women to come forward. These patients are afraid to come forward for several reasons including fear of retaliation from their partner and losing custody of their children [42,43]. There is a general mistrust of healthcare professionals in this population, due to fear of being judged, or even being blamed themselves for allowing someone to be violent towards them [4]. Usually, those who do come forward and seek help are doing so because the abuse has gotten severe enough that they are willing to risk the consequences of seeking help. Due to shame that surrounds both domestic violence and brain injuries, victims who suffer from TBI secondary to domestic violence often do not have the luxury of an early diagnosis. This makes this population unique from other TBI patient populations such as athletes, who usually get treatment right after their injury occurs [44].

Although early intervention has been proven to decrease negative outcomes in this population, research has shown that only 1 out of 10 physicians screened for domestic violence when interviewing patients. When questioned as to why, physicians cited lack of time, training, and resources as their main concerns. Physicians also stated that they had a fear of opening “Pandora’s box” because they were afraid that the patient would bring up a problem that they were unable to fix [33]. Even if physicians practice simple screening with questions such as “Do you feel safe at home?” or “Is anyone in your home hurting you?”, they oftentimes do not push for details, leading to missed diagnoses. There are minimal adverse effects of screening for TBI secondary to domestic violence [6]. The patient may be uncomfortable and may experience some emotional distress, but overall, it has been shown that screening had even greater benefits such as reducing the incidence of domestic violence and improving health outcomes in this population [45].

There are several individual screening tools available for TBI and domestic violence (Table 4), but there is still a need for a universal screening tool that encompasses questions addressing TBI due to domestic violence. In general, screening should consider the effects of both TBI and domestic violence. Patients may be experiencing elevations in symptoms such as fatigue, anxiety, and difficulty with concentration and memory [15]. Symptoms like these should not automatically be attributed to psychiatric issues and should instead be considered to be possible effects of the abuse and/or TBI. When screening patients, clinicians should avoid yes or no questions, and questions should be behavior centric. For example, patients should be asked if they have been hit before, not if they have experienced domestic violence [34]. Providers should pay close attention to note any inconsistencies in the patients’ stories [6]. A survey of domestic violence survivors showed that self-administered questionnaires were the preferred form of screening tool, and with this method, individuals were less likely to underreport the occurrence of domestic violence [42].

Regardless of which screening tool is used, it is important for all healthcare professionals to keep in mind that these patients need validation. The abuse should not be minimized, and the patients should not be stigmatized. It is also imperative that patients who screen positive for TBI secondary to domestic violence are provided with discharge instructions that are specific to their situation. Surveys of domestic violence survivors have shown that many times individuals were not able to access resources that addressed both TBI and domestic violence, so they often chose their safety over their healthcare [19]. It is not helpful for patients to be screened and diagnosed if they are not provided with resources that are relevant and beneficial to them. Since TBI and domestic violence are so closely linked in these patients, they will need more personalized resources than patients who only need resources for either TBI or domestic violence.

### 4.3. Neuroimaging Tools

When a patient presents with head injury, it is very common for the patient to receive neuroimaging in the form of a computed tomography (CT) scan. Unfortunately, CT scans are often not helpful in the diagnosis of patients with TBI secondary to domestic violence, as they are not able to detect abnormalities that result from mTBIs [48]. This is because the changes in the brain that occur from mTBI are often very subtle [3]. Despite this, imaging such as CT and magnetic resonance imaging (MRI) can be helpful in visualizing maxillofacial injuries, and any large brain contusions or bleeding that could be occurring because of the violence the patient has experienced [12]. It is also important to repeat neuroimaging with a CT scan in this patient population, as a primary TBI can worsen over time and there is always the risk of secondary injury occurring after the initial assessment [49].

More recently, a specific type of MRI called diffusion tensor imaging (DTI) has been used to detect disturbances in white matter structure due to TBI by assessing microstructural characteristics of the brain as well as the organization of fibers based on water diffusion. DTI uses fractional anisotropy (FA) to assess diffusion which can approximate axonal growth [50]. DTI can be used in clinic and may be beneficial in diagnosing domestic violence survivors with TBI, but its interpretation may still be challenging [51]. Overall, neuroimaging is an important part of diagnosing a patient, as it can allow for personalized evaluation and selection of treatments.

## 5. Treatment Models

There are many barriers to the care of patients who present with TBI secondary to domestic violence population. Due to their unstable social situations, it can be challenging for them to access proper communication and transportation methods [16]. Oftentimes, patients must choose between healthcare and maintaining their safety. Surveys have shown that many survivors have found that hospitals and clinics were not able to keep them safe, while shelters were not able to treat their head injuries. Many survivors face danger due to their immigration status, fear of further abuse, and the possibility of child protective services taking their children away [19].

The only true cure for traumatic brain injury is prevention [1]. Nevertheless, proper treatment can allow survivors to regain function, and prevent their injuries from worsening. When treating patients who have received a TBI secondary to domestic violence, it is imperative to act early and to take a multidisciplinary, team-based approach. Patients should be encouraged and referred to receive early treatment from a TBI specialist. These specialists should create a neurorehabilitation plan for patients that is specific to being a survivor of domestic violence [48]. Depending on the patient’s injury and its effects, the patient care team should also include a neurologist, social worker, psychiatrist, physical therapist, occupational therapist, and speech therapist [26]. All healthcare professionals should maintain contact with each other, as well as with social services and domestic violence professionals [3].

Patients should be referred to social services and domestic violence resources as soon as possible. It has been shown that an intervention within the first 24 h of the recognition of domestic violence can lead to the offender being arrested days earlier and lowers the probability of re-assault [34]. When creating a safety plan for patients, professionals should keep in mind that the effects from a TBI such as cognitive impairment and memory loss can make it difficult for patients to follow plans and care for themselves.

Overall, there is a significant need for more guidelines on treatments for patients who present with TBI secondary to domestic violence. Neurofeedback is an emerging treatment for symptoms of TBI secondary to domestic violence. Neurofeedback uses operant conditioning to regulate activity in various regions of the brain. In the past, it has been mostly used in veterans and survivors of addiction, but, recently, it has also been used on patients suffering from PTSD, post-concussive syndrome, and TBI. Results from neurofeedback treatment have shown increased calmness and decreased depression in PTSD patients, making it a viable treatment option for survivors of domestic violence [5].

## 6. Outcomes of TBI Due to Domestic Violence

Survivors of TBI due to domestic violence are more likely to have overall poor health outcomes, including both physical and mental health. The physical and mental toll of their previous trauma makes this population more likely to have worse health behaviors such as smoking tobacco, drinking alcohol, and using high quantities of pain medication [20]. Individuals in this population are also more likely to experience chronic pain, including neck and back pain [43]. Other long-term sequelae that these survivors tend to experience include an increase in gynecological, gastrointestinal, and central nervous system difficulties, as well as decreased immune function [25,50]. TBI due to domestic violence can have a significant impact on one’s mental health. Survivors from this population are more likely to suffer from anxiety, depression, and PTSD [13,52]. In addition, domestic violence survivors often experience emotional numbing and become avoidant, which can lead to abnormal relationship functioning [53]. TBI survivors may also experience increased aggression and could be at risk of becoming a perpetrator of domestic abuse [23].

Many neurological long-term consequences of TBI in this population may be due to second impact syndrome. These individuals often receive subsequent head injuries before full resolution of their first TBI, which can lead to fibrotic scarring in the brain [19,54]. This can cause dysfunctional connections between white matter in the brain which can make it difficult for patients to maintain efficient cognitive performance [19]. Chronic traumatic encephalopathy (CTE) is a long-term sequela of TBI secondary to domestic violence, which is a risk for many patients in this population.

Individuals who have experienced TBI from strangulation may experience outcomes secondary to hypoxia, such as seizures, coma, or even brain death [53]. Long-term neurologic outcomes of strangulation include decreased function in memory, learning, attention, and executive function [55]. Individuals who have suffered from TBI due to strangulation can suffer from aspiration pneumonia days to weeks after the event. This can lead to a progressive, irreversible encephalopathy [40].

### 6.1. PTSD

In women who have experienced domestic violence, 31–84% of them experienced long-term PTSD. PTSD prevalence in this population can vary, depending on where the sample is drawn, and which instruments are used to measure PTSD. PTSD can be caused by partner dominance, forced sex in a relationship, and physical violence. The severity of PTSD may increase with increased severity of physical violence as well as an increased mean number of traumas [26]. Individuals with PTSD may suffer from nightmares, making it difficult for them to receive adequate sleep [15,56]. In addition, mothers who experience PTSD because of domestic violence may have strained relationships with their children. A study showed that mothers who suffered from domestic violence PTSD had a distorted attachment to their children and, therefore, could have difficulty appraising their child’s emotional communication [57].

PTSD is typically treated with a combination of psychotherapy and medication. Battered women may respond best to cognitive behavioral therapy (CBT) that is specific for their situation such as cognitive trauma therapy for battered women (CTT-BW). CTT-BW includes psychoeducation, skills training, exposure to reminders of trauma, as well as assessment and correction of irrational beliefs. One study found that 12 out of 14 women who were treated with 90-min weekly sessions of CTT-BW had 50% symptom reduction. Neuroimaging with fMRI also showed reduced amygdala response which indicated enhanced processing of emotional salience as well as decreased fear learning [58].

### 6.2. Chronic Traumatic Encephalopathy

Chronic traumatic encephalopathy is an irreversible neurologic outcome associated with recurrent head impacts that lead to concussive and subconcussive injuries [59]. CTE was originally known as “punch-drunk disease” because it was first associated with boxers in the 1920s who would get hit in the head and then start acting drunk. Punch-drunk disease was first linked with battered woman syndrome in 1990, when a female patient who was chronically abused by her partner was found to have CTE post mortem [8]. Though CTE was found in this victim of IPV, there has not yet been enough research done on this population to directly link CTE as an outcome of IPV.

CTE can only be officially diagnosed upon post-mortem analysis. The pathognomonic lesions for this disease are perivascular, hyperphosphorylated tau-immunoreactive neurofibrillary tangles and abnormal neurites distributed at the sulcal depths [60,61]. Though it may not be officially diagnosed while patients are alive, there are clinical characteristics that when paired with a patient’s history, can strongly indicate CTE. CTE is a progressive disease, with most patients not showing symptoms until years or decades after their head injuries [59,60]. Patient with CTE may exhibit childish behavior, including violent and disinhibited outbursts. They may also experience depression, paranoia, and impulsivity. Individuals with CTE can appear as though they are under the influence of substances and/or that they are suffering from a psychiatric illness, and therefore, it is critical that these patients are evaluated with a thorough history, so that they are less likely to be misdiagnosed [59].

### 6.3. Risk to Future Generations

TBI and domestic violence both confer a risk to future generations. It has been shown that around 75% of women who had experienced TBI secondary to domestic violence had experienced abuse as a child [53]. The cycle typically starts with a child who is raised in a violent home and may obtain a TBI from an abusive family member during their childhood or adolescence. A family that experiences domestic violence likely has CPS involved at some point, which can lead to further instability in a child’s life and makes them more likely to enter an abusive relationship as an adult. The individual may then obtain a TBI from the abuse they experience at the hands of their partner. Multiple TBIs lead to severe consequences, which can make it difficult for these individuals to leave an abusive relationship. This means that their children will be raised in an unstable, abusive household, thereby, perpetuating the cycle. In addition, growing up in a violent family and having a TBI at a young age can also make a person more likely to become a perpetrator of family violence themselves [19].

The key to breaking this cycle is to get care and treatment for victims of TBI secondary to domestic violence as soon as possible. Several women who have experienced domestic violence have stated that they have experienced too many TBIs to count [62]. With each TBI, these women tend to have greater difficulty with memory, learning, focusing, and performing goal-directed behaviors [19]. In addition, patients may have difficulty relating to others, and can start to be very isolated [63]. This makes it nearly impossible for them to leave their situation without putting themselves and their children at serious risk. Additionally, living in an environment that leads to chronic stress makes it even more difficult to recover from a TBI [35]. For these reasons, women tend to stay in their abusive homes, which unfortunately means that their children are also stuck in this cycle [64]. By intervening early and providing these survivors with the proper resources, they may be able to leave the violence and alter their trajectory as well as that of their children’s lives.

## 7. Effects of COVID-19 on Domestic Violence

In general, there is a higher risk of domestic violence when there is increased stress on a relationship. The COVID-19 pandemic has caused increased physical and mental stress for many individuals, in particular, because it led to isolation during quarantine [65,66]. This isolation made it difficult for victims of domestic violence to access resources, including healthcare and shelters. The increased isolation also allowed for abusive partners to have increased control over their victims, without the risk of being caught, as staying inside made it less likely for someone to notice any physical injuries [66].

In addition to fewer individuals seeking assistance explicitly for domestic violence, individuals were also less likely to go to doctors’ offices for all reasons, which means that there was less general screening overall. This led to fewer incidental findings of domestic violence and its effects, and therefore less treatment. Suggestions for increasing the resources available to victims include adding routine IPV screenings wherever possible, including at COVID-19 testing and vaccination sites [66].

While it has been shown that there was an increase of reports and calls of domestic violence during the COVID-19 pandemic, there is still a need for more data about the impact of COVID-19 and its influence on the prevalence and outcomes of domestic violence.

## 8. Discussion

TBI secondary to domestic violence is an ongoing problem that affects millions of individuals each year. Despite the increase in research in recent years, there is still a dearth of research on this topic. TBI related to domestic violence needs to be studied longitudinally, and it needs to span more diverse populations. Populations such as people of color, the LGBTQIA+ communities, and those with disabilities are still not adequately represented in the research that has been done, making it difficult for them to receive proper treatment and care. It is also important to recognize that adolescents can also suffer from violence at the hands of an intimate partner, leading to TBI. There is also a need for future research on the connection between domestic violence and CTE. Overall, more education on this topic is needed. Increased education and awareness will allow providers to be more adept in recognizing signs and symptoms of TBI and domestic violence. Further educating patients and the general population is also necessary as it can lead to decreased discrimination and increased social support for individuals in these circumstances, making it easier for them to seek help and receive the care they need. Finally, it is imperative that a standardized screening tool is developed for TBI secondary to domestic violence. While there are separate screening tools for both separately, it is important for providers to have access to a standard tool that allows them to identify a possible diagnosis of TBI secondary to domestic violence. This screening tool should be paired with proper next steps if a patient screens positive, in order that patients are not being screened and left without any resources. TBI secondary to domestic violence is still a newly recognized issue, but continuing to increase education and awareness can provide patients in this population with a better chance of safety and recovery in their lifetimes.

## Figures and Tables

**Figure 1 brainsci-12-00122-f001:**
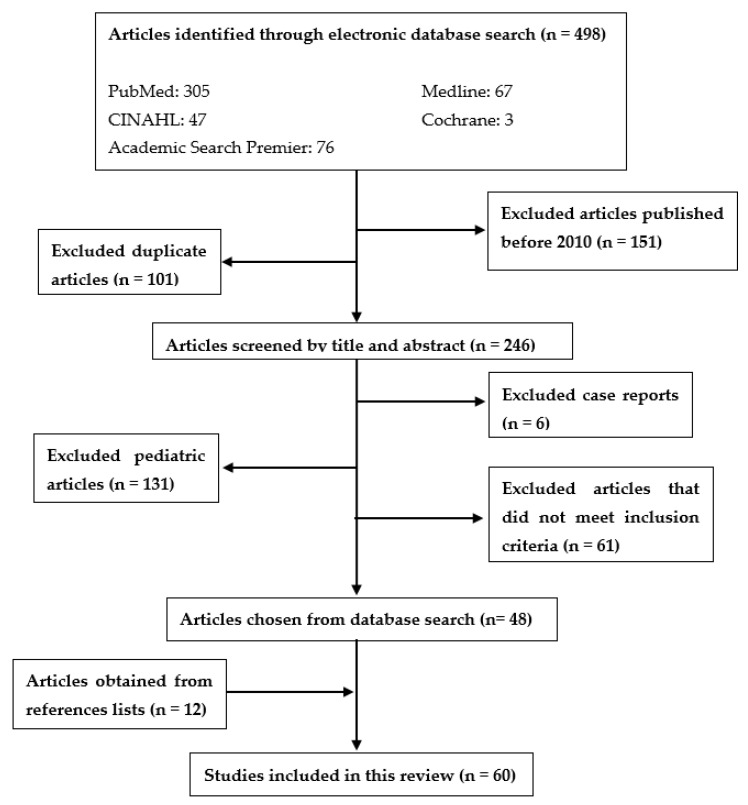
Method process flow chart.

**Table 1 brainsci-12-00122-t001:** Risk factors for TBI secondary to domestic violence.

Individuals at Increased Risk of TBI Secondary to Domestic Violence
Females of child-bearing age [12]
Individuals with disabilities [12]
Previous TBI (from domestic abuse or other cause) [2,3,15,16,17,18]
Low annual household income [11]
Minority races (Native American, Black, Asian) [2,11,19,20]
Female military veterans [21,22]
LGBTQIA+ communities [22]
Elderly (>60 years old) and children [13,23]

**Table 2 brainsci-12-00122-t002:** Measurement tools for domestic violence in the LGBTQIA+ communities.

Author	Screening Tool	Population	Measuring Tool	Study Objective	Conclusions
Dyar et al., 2021 [28]	Sexual and gender minorities-specific IPV Conflict Tactics Scale	Sexual and gender minority populations who were assigned female at birth between ages 16 and 20 years old.	An assessment scale addressing SGM-specific IPV behaviors including perpetriation and victimization.	To create a comprehensive set of culturally appropriate measures captivating a broad range of types of IPV among sexual and gender minorities.	This tool demonstrated high reliability and validity. It improved upon previous scales by being cuturally sensitive and addressing unique forms of IPV experienced by this community. This tool can be further improved and evaluated by future research and use on SGMs who are assigned male at birth, as well as different age groups.
Stephenson and Finneran, 2013 [29]	IPV Gay and Bisexual Men (-GBM) scale	Gay and bisexual men in the USA with at least a 6th grade reading level.	Consist of 23 prompts in five domains: physical and sexual abuse, monitoring, controlling, HIV-related abuse, and emotional abuse, which takes about 15–20 min to complete.	Create a new scale to measure IPV among gay and bisexual men.	The IPV-GBM scale showed strong internal reliablity. Although there was minimal variations in content of the scale by race, the results showed that it seemed to be appropriate for use in white and black/African American populations. This scale showed a higher prevalence of IPV than other scales, indicating that it included more items that are experienced specifically by this population. Further testing is needed to create a scale that is applicable to larger samples of this population as well as other racial/ethinic groups.
Peitzmeier et al., 2019 [30]	Transgender-related IPV Tool	Transmasculine individuals: individuals assigned a female sex at birth who have a non-female gender identity.	Four questions assessing four domains: coercive control of gender transition or gender presentation, emphasizing undesirability of transgender individuals as intimate partners, blackmail via outing, and sabotaging transition.	Create a measurement tool to assess transgender-related IPV.	This study showed that the scale had adequate reliability and validity as compared with other measures. This tool was helpful for identifying dangers that transgender individuals may be facing, that would be otherwise missed by traditional meaurement tools. A revised 10-item scale is being developed, as these additional items may increase the tools sensitivity and validity. Using this tool will allow clinicians and researchers to better identify IPV experienced by transgender individuals.

**Table 3 brainsci-12-00122-t003:** Common presenting symptoms of TBI secondary to domestic violence.

Common Presenting Symptoms of TBI Secondary to Domestic Violence
Neurologic symptoms: headache, confusion, memory loss [38]
Gastrointestinal symptoms: malnutrition, alcoholic cirrhosis, loss of appetite, abdominal pain [37,38]
Genitourinary symptoms (may indicate sexual abuse): vaginal infections, STDs, vaginal bleeding, pelvic pain, UTI [38]
Symptoms that may indicate strangulation: hoarseness, petechiae in conjunctivae, scalp, and external ear [40]
Psychiatric symptoms: emotionally labile, restless, combative, catatonic [40]
Other common symptoms: bruises, back pain, acquired thrombocytopenia, post-traumatic wound infection [26,37]

**Table 4 brainsci-12-00122-t004:** Screening tools that help identify TBI secondary to domestic violence.

Tool	Method of Administration	Primary Focus: TBI vs. IPV	Prompts Relating to TBI Secondary to IPV	Ways to Make the Tool Better at Screening for TBI Secondary to Domestic Violence
HELPS [2,42,46,47]	Professional interview	TBI	None, asks about injury to head, but not in relation to domestic violence.	When asking about injuries to the head, ask the cause. Specifically ask if the injury was inflicted by another individual and if so, who.
Brain Injury Screening Questionnaire (BISQ) [42]	Self-report	TBI	Asks about blows to the head.	Ask about the cause, i.e., have injuries ever been inflicted by their partner. Ask about other injuries to the face.
Ohio State University TBI Identification Method [42]	Interview	TBI	Addresses questions about TBI because of violent shaking.	Include prompts about blows to the face and head, ask if their partner has every physically abused them.
Traumatic Brain Injury Questionnaire [42]	Self-report	TBI	Asks about injury to face and injury secondary to cord around neck.	Further address injuries to the head and face, including blows to the head. Ask specifically if patient has ever acquired an injury from their partner.
Partner Violence Screen Questionnaire [12]	Interview	IPV	Have you been hit, kicked, punched, or otherwise hurt by somebody in the past year?	Follow up this prompt by asking about if there have been any injuries specifically to the head.
Hurt Insulted Threated or Screamed at Instrument [12]	Interview	IPV	How often does your partner physically hurt you?	Expand to ask about the type of physical abuse (blows, shaking, etc.) and if there has ever been an injury to the head.
Woman Abuse Screening Tool (WAST) [12]	Interview	IPV	Do arguments ever result in hitting, kicking, or pushing? Has your partner ever abused you physically?	Ask about body locations of physical abuse, including the head and face. Ask about timing of the most recent injury.

## Data Availability

Not Applicable.

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
