# Peer review of "Update on Domestic Violence and Traumatic Brain Injury: A Narrative Review"

_brainsci, 2022, doi:10.3390/brainsci12010122_

Round 1

Reviewer 1 Report

The authors have done an excellent job of outlining the current literature of TBI as a result of IPV.  The tables are a helpful representation of the literature (Figure 1) and the risk factors, symptomatology, and assessment tools.  I do have several comments:

 1) Page 3 the authors cite one reference regarding disability. Please check Ballan et al who have done extensive work on IPV, TBI and disability.           

2) Table 2, author Peitzmeier et al., there is a typo in the “study objective” column. 

3) On page 10, the authors name quite a few professionals that can intervene with an IPV survivor who has a TBI, yet they do not mention the most significant front-line worker: social workers.  Social workers will be the first healthcare professional to come in contact with an IPV survivor (they populate most of the 1,900 DV shelters nationwide).

3) Page 13, 6.2, While we strongly suspect that Chronic Traumatic Encephalopathy (CTE) will be widely observed in the IPV population, I am not aware of any studies (Other than the Roberts early observation in the 70s) observing CTE in an IPV population.  If the authors know of such a reference or study, please state it. Otherwise, state that this has yet to be borne out with this population.

Author Response

Reviewer 1,

We greatly appreciate your comments on our paper and agree that these changes will enhance our review article.  We have taken your comments into consideration and have made the following changes to our manuscript: 

  1. Page 3 now includes Ballan as a reference regarding the increased risk of IPV in disabled populations
  2. The typo in the Table 2 “study objective” column has been corrected.
  3. Social workers are now added as a professional that should be involved in the care team of TBI due to IPV patients, as they do play a significant part in the care and management of this population.
  4. It has been noted that while CTE may have been noted in an individual woman who experienced domestic violence, CTE as an outcome of IPV has still yet to be concretely linked to this population.

We again thank you for your time and your comments on our paper.

Reviewer 2 Report

  1. Perhaps this journal does not require to have heading such as background, method, results, conclusion etc. for the Abstratc. However, please consider providing information regarding background, method, results, conclusion etc.
  2. Probably it is too ambitious to state that a literature review can improve the diagnosis, evaluation, and treatment of TBI. Direction for future research can be the only objective from a narative literature review.  

Author Response

Reviewer 2,

We greatly appreciate your comments on our paper and agree that these changes will enhance our review article.  We have taken your comments into consideration and have made the following changes to our manuscript: 

  1. The abstract has been edited to include background, method, results, and conclusion
  2. The abstract has been reworded to indicate that the goal of this article is to improve future research on this topic.

We again thank you for your time and your comments on our paper.